# Label-Free Characterization of Macrophage Polarization Using Raman Spectroscopy ^[note 1]^

**DOI:** 10.3390/ijms24010824

**Published:** 2023-01-03

**Authors:** Max Naumann, Natalie Arend, Rustam R. Guliev, Christian Kretzer, Ignacio Rubio, Oliver Werz, Ute Neugebauer

**Affiliations:** 1Leibniz Institute of Photonic Technology Jena, Member of Leibniz Health Technologies, Member of the Leibniz Centre for Photonics in Infection Research, LPI, Albert-Einstein-Str. 9, 07745 Jena, Germany; 2Center for Sepsis Control and Care, Jena University Hospital, Am Klinikum 1, 07747 Jena, Germany; 3Department of Pharmaceutical/Medicinal Chemistry, Institute of Pharmacy, Friedrich Schiller University Jena, Philosophenweg 14, 07743 Jena, Germany; 4Department of Anaesthesiology and Intensive Care Medicine, Jena University Hospital, Am Klinikum 1, 07747 Jena, Germany; 5Jena Center for Soft Matter (JCSM), Friedrich Schiller University Jena, Philosophenweg 7, 07743 Jena, Germany; 6Institute of Physical Chemistry and Abbe Center of Photonics, Friedrich Schiller University, Helmholtzweg 4, 07743 Jena, Germany

**Keywords:** macrophage phenotype, Raman spectroscopic imaging, chemometric unmixing, principal component analysis and linear discriminant analysis (PCA-LDA)

## Abstract

Macrophages are important cells of the innate immune system that play many different roles in host defense, a fact that is reflected by their polarization into many distinct subtypes. Depending on their function and phenotype, macrophages can be grossly classified into classically activated macrophages (pro-inflammatory M1 cells), alternatively activated macrophages (anti-inflammatory M2 cells), and non-activated cells (resting M0 cells). A fast, label-free and non-destructive characterization of macrophage phenotypes could be of importance for studying the contribution of the various subtypes to numerous pathologies. In this work, single cell Raman spectroscopic imaging was applied to visualize the characteristic phenotype as well as to discriminate between different human macrophage phenotypes without any label and in a non-destructive manner. Macrophages were derived by differentiation of peripheral blood monocytes of human healthy donors and differently treated to yield M0, M1 and M2 phenotypes, as confirmed by marker analysis using flow cytometry and fluorescence imaging. Raman images of chemically fixed cells of those three macrophage phenotypes were processed using chemometric methods of unmixing (N-FINDR) and discrimination (PCA-LDA). The discrimination models were validated using leave-one donor-out cross-validation. The results show that Raman imaging is able to discriminate between pro- and anti-inflammatory macrophage phenotypes with high accuracy in a non-invasive, non-destructive and label-free manner. The spectral differences observed can be explained by the biochemical characteristics of the different phenotypes.

## 1. Introduction

Macrophages are important cells of the innate immune system. They are present in essentially all tissues and play distinct roles in the initiation and resolution of an inflammatory response. Macrophages are involved in a variety of tasks, such as fighting in the frontline against invading pathogens via phagocytosis or pathogen killing, but also signaling and cross-talk with other immune cells [1]. Macrophages’ complex functional variability and adaptability to different infectious situations is based on their extensive phenotypic plasticity. Tissue-resident, naïve macrophages (M0), but also circulating monocytes can be recruited for differentiation into macrophages and further for polarization into numerous functional subsets of macrophages. Polarization is accurately regulated by exogenous and intrinsic inflammatory stimuli and accompanied by morphological changes as well as intracellular alterations in the transcriptome, proteome and metabolome. In traditional classification models, these polarized macrophages are described as classically-activated, pro-inflammatory M1 or alternatively-activated, anti-inflammatory M2 macrophages [2,3]. Although, recent research has uncovered the existence of numerous other subtypes, this traditional classification is still considered valid and is often used for simplicity’s sake. Pro-inflammatory M1 macrophages are usually induced following the encounter with pathogen-associated molecular patterns (PAMPs) such as lipopolysaccharide (LPS), a bacterial cell wall constituent, or inflammatory mediators, e.g., interferon γ (IFN-γ). Such classical activation provokes an enhanced production of further pro-inflammatory factors by the macrophage like IFN- γ, TNFα, IL-1β and further cytokines and chemokines, contributing to the progression of the inflammatory host response. In contrast, an alternatively pattern of stimuli such as interleukin 4 (IL-4) causes the polarization towards anti-inflammatory M2-like macrophages. This macrophage phenotype is involved in tissue repair and remodeling processes and other processes contributing mainly to the resolution of inflammation and regeneration [4,5,6,7,8]. Importantly, disturbances in controlled macrophage polarization may ultimately lead to maladaptive immune responses that cause severe tissue damage, fibrosis and cancer, or contribute to severe inflammatory conditions such as sepsis [9].

The assignment of isolated cells to a macrophage subtype is usually based on morphological assessment by (fluorescence) microscopy or fluorescence marker-based phenotyping methods like flow cytometry. For the latter, differential expression levels of surface activation markers, such as e.g., CD11b, CD54, CD80, CD163 and CD206, are used to differentiate the phenotypes [10,11]. In situ or even in vivo, i.e., directly within intact tissues samples, those methods are not easily applicable. In addition, tissue-resident macrophages were reported to form complex or even hybrid functional phenotypes rather than distinct and stable phenotypic subsets [6]. Therefore, alternative, and ideally label-free methods, for characterizing the phenotype of macrophages would be desirable. Raman spectroscopy was found in recent years to be a powerful tool for cell characterization and cell-type identification as the Raman spectroscopic fingerprint reflects the overall chemical composition, including proteomic and metabolomic features [12,13]. This information can, moreover, be collected in a spatially-resolved manner resulting in false color images without the need of any dyes or labels [14,15]. The technique has been successfully applied to distinguish immune cell subtypes [16,17,18,19,20], but also different cellular activation states after in vitro [21,22,23] and in vivo stimulation [24].

In the present study, we explore Raman spectroscopic imaging as a cell-biological tool for the characterization and distinction of different functional macrophage phenotypes induced in vitro. Macrophages were derived by differentiation of primary human monocytes and polarization to the various phenotypic subsets was ascertained with established reference methods, i.e., flow cytometry-based phenotyping and the evaluation of morphological features using confocal laser scanning microscopy (cLSM). 

## 2. Results and Discussion

### 2.1. Verification of Macrophage Polarization by Marker Phenotyping

Successful differentiation of macrophages from isolated human monocytes and polarization into pro- or anti-inflammatory phenotypes was verified by flow cytometry. To this end, macrophages had to be brought in suspension after polarization. While M0- and M2-like macrophages were strongly adherent to the well bottom, M1-like cells were only loosely attached or already in suspension. Detached M0 and M2 macrophages showed a broader distribution in forward and side scatter than M1 macrophages, indicative of a more homogenous distribution in size and granularity of M1 cells (Appendix A). CD11b (integrin-α_M protein subunit) as a general surface marker for macrophages, CD80 (B7-1 membrane protein) as marker for pro-inflammatory M1 macrophages, and the mannose receptor CD206 as surface marker for anti-inflammatory M2 macrophages were used for quantitative assessment of the differentiation efficiency (Figure 1). 

As expected, the general macrophage surface marker CD11b was found on 79 ± 3% of cells in M0 macrophage samples, validating the formation of this non-activated phenotype from isolated monocytes after stimulation with GM-CSF and M-CSF (Figure 1A,C). Besides the observation of a potentially unspecific signal in the isotype control of donor 1, all results appeared to be consistent among the three donors. Analysis of the surface markers CD80 and CD206 in our samples (Figure 1B,C) revealed that CD80 was significantly (*p* ≤ 0.05) more abundant on pro-inflammatory M1 macrophages (59 ± 5 %) compared to M2 macrophages (14 ± 13%). CD206 vice versa had a significantly (*p* ≤ 0.01) increased expression level on anti-inflammatory M2 macrophages (86 ± 9%) compared to M1 macrophages (50 ± 7%). This proves the successful formation of the respective macrophage phenotype. It is known that the surface markers CD80 and CD206 are not exclusively expressed on pro- and anti-inflammatory M1 and M2 macrophages, respectively, but rather co-expressed with different ratios [5] as is also found in our study. 

The confocal laser scanning microscopy (cLSM) of adherent cells revealed the characteristic size and shape of the macrophages (Figure 2, Appendix A) [25]. Pro-inflammatory macrophages showed a compact round shape and were filled with lipid droplets (Figure 2B,E), while resting and anti-inflammatory macrophages were flat, elongated with partially stretched extensions (Figure 2A,C,D,F). Actin-rich sites at the periphery (bright dots in Figure 2A,C) indicated the presence of podosomes. On the opposite cellular terminal of the podosomes, filopodia, thin cytoplasmic projections, were visible (Figure 2A,C).

In summary, both fluorescence-based techniques (flow cytometry as well as cLSM) could confirm differentiation into the main phenotypes M1 and M2. However, subtype populations were not 100% pure, as divergent phenotypes were also observed, e.g., small round cells in M2-differentiated samples (Figure 2C).

### 2.2. Raman Spectroscopic Imaging Reveals Chemical Differences in a Spatially Resolved Manner

Raman image scans from 65 primary, chemically fixed, human monocyte-derived macrophages were analyzed using the spectral unmixing algorithm N-FINDR to visualize the chemical composition of the different phenotypes in a label-free and non-destructive manner. Different Raman spectra were assigned to the chemical component lipids, proteins and nucleic acids; low intensity spectra with spectral features of organic components are merged into the class cytoplasm/environment (Appendix A). The spatial abundance of those components is visualized in false-color Raman images in Figure 2G–I and Appendix A. Those images show high similarities to the fluorescence images in Figure 2A–F. However, it is important to note that false-color Raman images are recorded non-invasively, without probes or stains and thus depict the intrinsic chemical contrast of the cells. The round silhouette of pro-inflammatory M1 macrophages and their high content of lipid droplets is ostensible, as well as the larger, extended shape of naïve and anti-inflammatory macrophages. Cell nuclei can be spotted and visualized based on the specific Raman bands of nucleic acids. 

Automated quantitative analysis of the false-color Raman images was performed to analyze differences in the chemical composition of the macrophage phenotypes. The distribution of the endmember coefficients for the four endmembers (lipids, proteins, nucleic acids, and environment) in the different cells is depicted in Appendix A. The relative distribution of proteins and nucleic acids did not show consistent differences from donor to donor and thus will be not discussed further at this point. However, in cells from all three donors, the contribution of lipids in pro-inflammatory M1 macrophages was significantly higher than in the other two phenotypes (*p*-values: 0.0047 for Donor 1, 0.099 for Donor 2, and 0.00006 for Donor 3), indicating a higher abundance of lipids in classically activated M1 macrophages. This notion was in agreement with the microscopic fluorescence images as well as Raman false color images (Figure 2D–I). Raman spectra of the lipids present in the different macrophage phenotypes were computationally extracted from the Raman maps using the relative abundance of the lipid endmember. Those spectra are presented in Appendix A together with details of extraction in the caption. Spectral differences between the lipid spectra from the different macrophage phenotypes point to a slightly different lipid composition. The Raman band ratio of the prominent C=C stretching band around 1655 cm^−1^, and the CH_2_ stretching band around 1444 cm^−1^ can be used to estimate the degree of unsaturation of fatty acids or other lipids by analyzing the abundance of n(C=C)/n(CH_2_) [26]. The ratio of the integral intensities of the Raman bands at 1655/1444 cm^−1^ was determined to be 1.60 (±0.16) for M0, 1.41 (±0.20) for M1 and 1.69 (±0.13) for M2 macrophages (Appendix A). These ratios indicate that among the many different lipids present in M1 macrophages the relative abundance of unsaturated fatty acids and lipids is slightly lower as compared to the relative composition of fatty acids and lipids in M0 and M2 macrophages. This is in agreement with previous findings from lipidomic and other approaches [27], showing that saturated fatty acids and oxylipins strongly contribute to type 1 inflammation.

Another Raman spectral difference between lipids found in classically activated M1 macrophages, M0 and M2 macrophages is the ratio of relative intensities of the Raman bands at 1260 and 1300 cm^−1^. Using typical spectral features of lipids, the slightly higher relative intensity of the Raman band at 1260 cm^−1^ is found in linoleic and α-linoleic acid as well as in the respective triacylglycerides trilinolein and trilinolenin [28], pointing to a higher relative abundance of those fatty acids in triacylglycerols in M1 macrophages. Lipids from alternatively activated M2 macrophages have a lower Raman band of the ester vibration around 1740 cm^−1^ (Appendix A) pointing to a lower relative content of triacylglycerols, but higher content of fatty acids.

The observed difference in the Raman spectra corresponds well with the known biochemical differences in the macrophage phenotypes. While LPS/IFNγ promotes fatty acid-generating pathways like anabolic fatty acid synthesis in M1 macrophages, M2 macrophages are known to have an increased uptake and catabolism of fatty acid via catabolic fatty acid oxidation or β-oxidation [29,30]. Previous mass spectrometry-based targeted lipidomic profiling of (bone-marrow-derived) unpolarized M0 macrophages, LPS + IFNγ-stimulated M1 and IL-4-stimulated M2 macrophages did also illustrate increased triglyceride levels in M1 macrophages as well as a shift in compositional profile toward polyunsaturated fatty acid–containing triglycerides [31].

### 2.3. Differentiation of Macrophage Phenotype Using Averaged Raman

In order to explore if the Raman spectroscopic fingerprint of the cells can be used as a tool to distinguish and characterize the different in vitro activated macrophage phenotypes, pre-processed spectra were averaged per each image. Figure 3 shows the averaged spectra for each donor. Characteristic Raman bands of macrophages can be detected and agree with published Raman data [22,32]: amide I band around 1650 cm^−1^ indicating proteins, C-H deformation band around 1441 cm^−1^, amide III band vibrations in the spectral region between 1230 and 1300 cm^−1^.

The most dominant differences occur between pro-inflammatory M1 and the other two phenotypes (naïve M0 and anti-inflammatory M2), predominantly in the spectral regions of lipids as discussed above (CH, CH_2_ and CH_3_ stretching and deformation regions at ~1300 cm^−1^, ~1444 cm^−1^). Importantly, those differences are consistent from donor to donor. The Raman spectra of monocyte-derived M0 macrophages and alternatively activated M2 macrophages are very similar. Characteristic differences might be hidden in the low wavenumber spectral region between 600 and 900 cm^−1^, (Figure 3: enlarged insets), yet they are not consistent between individual cells and between the donors. More data from more donors are necessary to draw reliable conclusions here.

The presence of consistent spectral differences between pro-inflammatory M1 macrophages and M0/M2 phenotypes encouraged us to compute PCA-LDA models for automated discrimination of M1 macrophages from other macrophage subsets. The model was rigorously controlled for overfitting: it was cross-validated by a leave-one donor-out strategy and the variation of quality metrics and LDA coefficients during the cross-validation process was carefully monitored. Appendix A provides detailed information on variation of quality metrics and LDA coefficients. Two principal components were found to be sufficient for a robust classification model reaching balanced accuracy of 86%, sensitivity 93%, specificity 85% and area under ROC curve (ROC AUC) 93% (calculated on combined testing predictions of the cross-validation). The 2D-PCA scores plot of the first two principal components is provided in Figure 4A. The PCA-LDA coefficients of this final classification model are shown in Figure 4B and reveal clear spectral features, which are very similar to a typical lipid spectrum displayed in Appendix A As the PCA-LDA coefficients reflect spectral differences responsible for discrimination of M1 and M0/M2 macrophage phenotypes, we conclude that the higher abundance of lipids in the classically activated M1 macrophages is mainly responsible for the spectral differentiation if averaged Raman spectra per cell are used. This is agreement with the analysis of the false-color Raman images and the higher abundance of lipid endmember spectra in M1 as compared to M0 and M2 cells, as discussed above (Appendix A). Training a similar classification using individual spectra (i.e., without averaging the spectra per image) leads to an accuracy of 70% with approximately similar PCA-LDA coefficients (Appendix A). Altogether, the results of PCA-LDA analysis suggest that lipids are a highly suitable factor for the identification of the pro-inflammatory M1 phenotype. 

Similarly, a PCA-LDA model discriminating between naïve M0 and anti-inflammatory M2 phenotypes was trained. Using cross-validation, the accuracy of the model was only around 50% (Appendix A). However, a more detailed look at the spectra shows that there are some donor-dependent differences, mostly in the spectral region 600–900 cm^–1^ (Figure 3). It is therefore, expected that the inclusion of more donors could reveal characteristic differences also between the closely related M0 and M2 phenotypes. 

## 3. Materials and Methods

### 3.1. Cell Isolation and Differentiation to Macrophage Phenotypes

Leukocyte concentrates of peripheral blood from three human healthy donors without infectious episodes and/or anti-inflammatory treatment for the last 10 days were obtained from the Institute of Transfusion Medicine of the Jena University Hospital. Protocols concerning sample collection, isolation and cell culture were approved by the ethics committee of the Jena University Hospital, ethical vote 5050-01/17. All methods are in accordance with the relevant regulations and guidelines. Monocytes were isolated from leukocyte concentrates by applying dextran-PBS (2.5% *v*/*v* dextran from *Leuconostoc* spp., MW ~40,000, Sigma Aldrich, Munich, Germany) followed by density centrifugation with leukocyte separation medium Histopaque^®^-1077 according to standard procedures (Sigma Aldrich). Peripheral blood mononuclear cells (PBMCs) were collected from the top of the gradient, washed several times with ice-cold PBS and seeded in cell culture flasks for 1h (37 °C, 5% CO_2_) in PBS (containing Ca^2+^/Mg^2+^) to isolate the adherent monocytes. For differentiation and polarization, established protocols were used [33]. Briefly, the non-activated phenotype M0 was generated by treatment of monocytes with granulocyte macrophage-colony stimulating factor (GM-CSF, Peprotech, Hamburg, Germany) and macrophage-colony stimulating factor (M-CSF, Peprotech) for 6 days. Inflammatory (M1) macrophages were obtained by treatment with 20 ng/mL GM-CSF for six days in 10% fetal bovine serum (Sigma Aldrich), 2 mM l-glutamine (Sigma Aldrich) and penicillin-streptomycin supplemented RPMI 1640 (Sigma Aldrich), and subsequent activation with 100 ng/mL lipopolysaccharide (LPS, Sigma Aldrich) and 20 ng/mL interferon- γ (IFN-γ, Peprotech). To obtain anti-inflammatory M2 macrophages, isolated monocytes were treated with 20 ng/mL M-CSF for six days in RPMI 1640 supplemented with 10% fetal bovine serum, 2 mM l-glutamine and penicillin-streptomycin, and subsequently activated with 20 ng/mL interleukin-4 (IL-4, Peprotech) for 48 h. [33] Approximately 1 × 10^6^ cells in 1 mL of each macrophage phenotype were plated in 12-well plates for flow cytometry, coated onto glass slides for fluorescence microscopy or on CaF_2_-slides for Raman spectroscopic imaging.

### 3.2. Fluorescence Staining, Flow Cytometry and Confocal Laser Scanning Microscopy

Samples of the three characteristic macrophage phenotypes (M0, M1, M2) were chemically fixed in 4% methanol-free paraformaldehyde (PFA, *m*/*m* in 1× PBS, pH = 7.4; Sigma-Aldrich, Hamburg, Germany). Macrophages were stained with phenotype-specific, fluorescence-labelled antibodies (2% *v*/*v* of M0: CD11b AB [anti-human, PE], M1: CD80 AB [anti-human, PE-VIO 770] and M2: CD206 AB [anti-human/mouse, PE]; Miltenyi Biotec, Bergisch Gladbach, Germany) and analyzed by flow cytometry on a BD Accuri^TM^ C6 Plus Flow Cytometer. Cells stained with either mouse IgG1 with PE/PE-VIO770 or rat IgG2b with PE labelling served as isotype controls (IC). Flow cytometry data were analyzed using the software FlowJo^TM^ 10.

For confocal Laser Scanning Microscopy (cLSM), macrophages were stained with phalloidin-AF555 (3% *v*/*v*; Thermo Fisher Scientific, Darmstadt, Germany)-DAPI (0.1% *v*/*v*)-staining (Merck, Darmstadt, Germany). Samples of the third donor were additionally double stained with Nile Red (1 µg/mL; Sigma-Aldrich, Hamburg, Germany) and DAPI. Staining procedures were performed at room temperature.

In order to compare the morphology of macrophage phenotypes, images of fixed cells in the M0, M1 and M2 states were acquired using the Axio Observer Z1/7 CLSM at room temperature and a Plan-Apochromat 20×/0.8 or a Plan-Apochromat 63×/1.4 oil immersion objective. Excitation and emission wavelengths were set to 553 nm/568 nm and 353 nm/465 nm for Phalloidin and DAPI, respectively. ZEN 3.2 (blue edition) and ImageJ bundled with 64-bit Java 1.8.0_172 were used for image processing. The latter included histogram equalization for each image color channel.

### 3.3. Raman Spectroscopic Imaging of Different Macrophage Phenotypes

To assure reproducibility, also if measurements were not performed immediately and on the same day, cells were chemically fixed prior to analysis in this study.

Following cell fixation with 1 mL Roti^®^ Histofix (Carl Roth, Karlsruhe, Germany) for 10–20 min at room temperature on CaF_2_-slides (Crystal GmbH, Berlin, Germany), the samples were washed twice with 1 mL 1× PBS and stored in 1× PBS at 4 °C until the measurements.

For Raman spectroscopic imaging a α 300 Raman spectrometer (WITec, Ulm, Germany) equipped with a 785 nm laser (75 mW in the object plane; Toptica GmbH, Munich, Germany) and a 60×/1.0 NIR Apo water immersion objective (Nikon GmbH, Düsseldorf, Germany) was used. An optical fiber (Ø 100 μm) was applied to guide the backscattered Raman light to a spectrometer. The signal was detected by a back-illuminated deep depletion CCD camera (DV401A-BV-352 cooled to −60 °C, Andor, Belfast, Ireland). For imaging of macrophages, a spatial step size of 0.5 µm and an integration time of 1 s per spectrum was chosen. In total, 67 cells (15 M0, 32 M1 and 20 M2) were measured yielding in 506,246 spectra (Detailed split per donor is given in Appendix A).

### 3.4. Raman Data Analysis

Raman images were recorded to capture the full cell, thus also non-cellular background was present in the images. To remove this background, images were masked by manual selection of the cellular shape using the Image Segmenter App in MATLAB R2020a. Further pre-processing and analysis of the Raman spectra was done using R (version 4.1) [34], R Studio (version 1.4) IDE and the following packages. ggplot2 [35], imager [36], gridExtra [37] for visualizations, MASS [38] for LDA calculation, hyperSpec [39] and dplyr [40] for data import and manipulation, matrixStats [41] for optimized matrix operations, and unmixR [42] for N-FINDR algorithm. 

Outlier spectra within the cells exhibiting too high fluorescence background were excluded, cosmic ray noise was automatically corrected by algorithm described in Ryabchykov, O. et al. [43]. The silent region 1800–2650 cm^−1^ was cut out and baseline was corrected by applying SNIP [44] algorithm to the spectra regions 350–1800 cm^−1^ and 2650–3100 cm^−1^ separately using 200 and 50 iterations, respectively. After re-processing, 65 Raman images with 183,596 spectra were kept for further data analysis (Appendix A).

To generate false color Raman images, N-FINDR algorithm [45] was used for linear spectral unmixing. Assuming that each spectrum is a mixture (linear combination) of some pure component spectra unknown beforehand (called endmembers), identification of those pure component spectra with following calculation of respective concentrations (non-negative coefficients; also called abundances) is called spectral unmixing. N-FINDR is a commonly used algorithm for finding the endmember spectra. In simple words, it can be said that N-FINDR algorithm searches for *p* most different spectra in the data, so that the rest of the data can be presented as non-negative linear combinations of those spectra. Assuming that for each component there is at least one pure endmember spectrum in the data, it is based on the fact that in N-dimensional space a simplex formed of the purest pixels (i.e., endmembers) has the largest volume (comparing to simplexes formed of non-pure pixels). More details on the algorithm and its implementation can be found in the original work [45] and in the R package [42].

Initially, the N-FINDR analysis was applied for each Raman image separately. Then 20 images (7 M0, 6 M1, 7 M2) were selected for combined analysis. The selection was based on the quality of the cell structure (nucleus, lipid droplets, etc.) resolution and the balance in number of cells of each phenotype. For these 20 images, common endmembers were calculated, i.e., N-FINDR algorithm was run on all chosen images combined. Finally, the obtained endmembers were applied for unmixing all 65 images. The number of endmembers was chosen plentifully to avoid influence of outlier spectra. Therefore, 15 endmembers were calculated and then manually, based on their spectral features, assigned to lipids, proteins, nuclei, and environ, and then grouped according to the assignment (Appendix A).

PCA-LDA, i.e., Principal Component Analysis (PCA) [46] followed by Linear Discriminant Analysis (LDA) [47,48], was used for building discrimination models for the phenotypes. We have selected this supervised analysis method as it had proven its potential in many classification problems using spectral data [49]. First, a PCA-LDA model discriminating between pro-inflammatory (M1) and not pro-inflammatory (i.e., M0 and M2 combined) was built. Then, a second model, discriminating M0 and M2 phenotypes, was set up. The discrimination models were calculated by using one average spectrum per image. Before taking the average, all spectra were mean normalized. Building a model on individual spectra level was also performed, however that did not improve discrimination quality (Appendix A).

While training PCA-LDA models, avoiding overfitting was controlled in different ways. All discrimination models were trained using leave-one donor-out cross-validation (i.e., the model was trained using only two donors and then applied to the third donor for testing; and the same was repeated for each donor). Principal components were calculated using only training data sets. Accuracy metrics for a model were estimated using only testing predictions. The number of principal components was chosen by considering both variation of quality metrics and variation of PCA-LDA coefficients during cross-validation iterations (Appendix A).

## 4. Conclusions

The study using Raman imaging of macrophages derived from primary human monocytes confirmed the potential of Raman spectroscopy as a non-invasive biophotonic tool for classification and characterization of different functional macrophage phenotypes. Single cell Raman spectroscopy was able to capture characteristic features of monocyte-derived naïve M0, classically-activated M1 and alternatively-activated M2 macrophages. This included in vitro cellular morphology as well as spectral characteristics. False-color Raman images that were obtained without any labels showed high similarity to images from cLSM obtained from cells after fluorescence-staining. 

Raman spectral characteristics of individual macrophages were used to train a reliable classification model to distinguish between pro-inflammatory M1 macrophages and M0/M2 macrophages with a robust accuracy of 85–90%. The increased lipid content found in M1 macrophages likely contributes to the cellular differentiation process. However, the current sample size of 65 cells (183,596 spectra after pre-processing) from three different donors was insufficient to achieve a discrimination of the highly similar M0 and M2 phenotypes using the Raman spectroscopic fingerprints. 

In our study, we chemically fixed the cells prior to Raman analysis. However, Raman analysis can also be done directly on living cells [50,51]. Requiring hardly any consumable costs and laboratory sample preparation, our findings underscore that Raman spectroscopy is a promising future diagnostic tool in clinics for characterization of the host immune response.

## Figures and Tables

**Figure 1 ijms-24-00824-f001:**
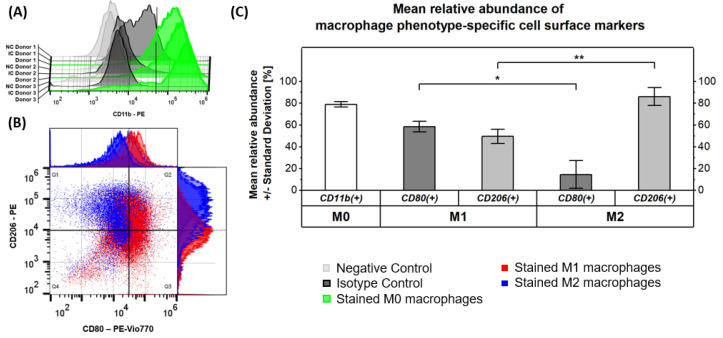
Validation of macrophage differentiation and activation by flow cytometric analysis of macrophage phenotype-specific cell surface markers. (**A**) The graph validates the formation of M0 macrophages from isolated monocytes, stimulated with GM-CSF and M-CSF by a high abundance of CD11b^+^ cells within the M0 macrophage samples originating from three donors (Donor 1 = solid line, Donor 2 = dashed line, Donor 3 = dotted line). (**B**) Successful activation of functional macrophage phenotypes after addition of LPS/IFNy or IL-10 is confirmed by the scatter plot of double-stained M1 and M2 macrophage samples from three donors. (**C**) The graph summarizes the mean relative abundance of macrophage phenotype-specific cell surface markers and indicates significant differences (* *p* ≤ 0.05, ** *p* ≤ 0.01).

**Figure 2 ijms-24-00824-f002:**
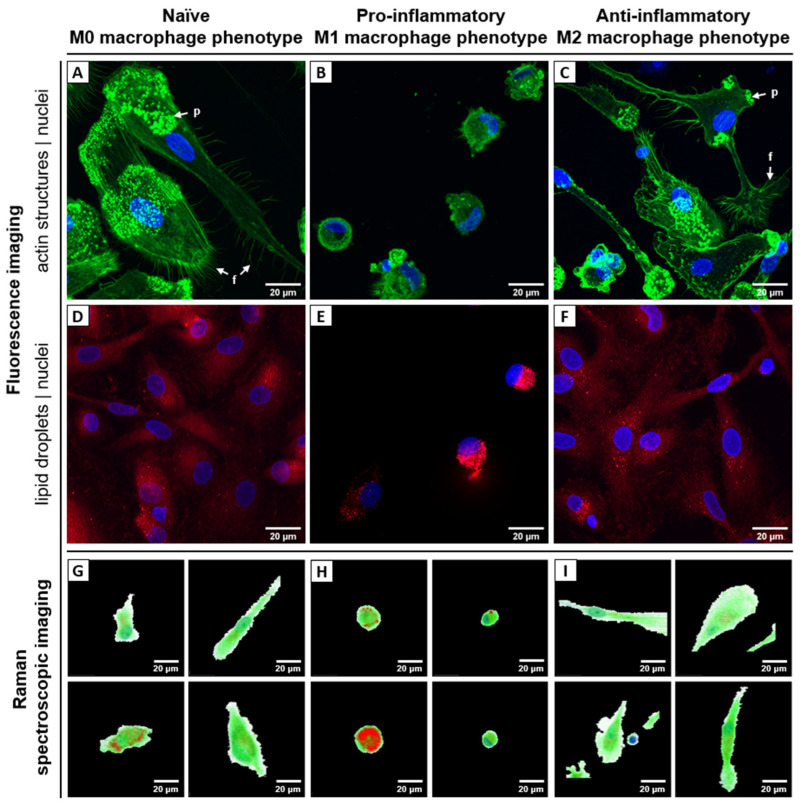
Biophotonic visualization of different human macrophage phenotypes: (**A**–**F**) fluorescence images, (**G**–**I**) Raman false-color images. Left column (**A**,**D**,**G**) shows naïve resting macrophages (M0) derived from blood monocytes after differentiation with GM-CSF/M-CSF, middle column (**B**,**E**,**H**) shows pro-inflammatory M1 macrophages activated with GM-CSF/ LPS/ INF-γ, and right column (**C**,**F**,**I**) shows anti-inflammatory M2 macrophages activated with M-CSF/ IL-4. (**A**–**C**) Fluorescence images. Cell nuclei are stained with DAPI (blue) and filamentous actin structures of the cytoskeleton with Phalloidin-AlexaFluor555 (green). Arrows indicate filopodia (f) and podosomes (p). (**D**–**F**) Fluorescence images. Cell nuclei are stained with DAPI (blue) and intracellular lipid droplets with Nile Red (red). (**G**–**I**) False color Raman images of different individual cells generated by N-FINDR analysis. The pixels are colored according to contribution of corresponding endmember group: red—lipids, green—proteins, blue—nuclei, white—environment. Corresponding brightfield images of the cells depicted in panel G-I can be found in Appendix A. Endmember spectra as well as further examples of false-color Raman images are presented in Appendix A. Scale bar: 20 µm.

**Figure 3 ijms-24-00824-f003:**
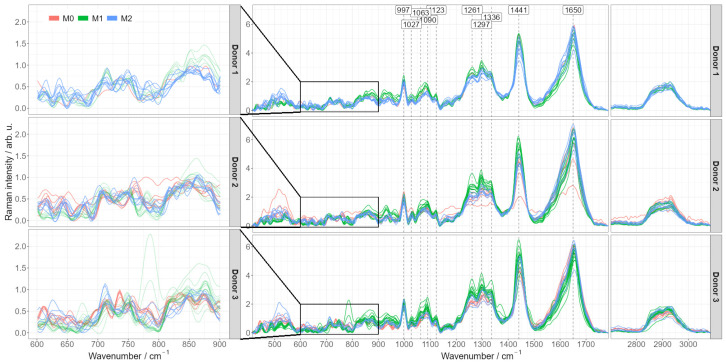
Pre-processed mean Raman spectra of different activated macrophage phenotypes split by donors. Pre-processed Raman mean spectra colored according to their specific phenotype (M0, M1 or M2) and grouped by the donor. The figures on the left provide a detailed view of the 600–900 cm^−1^ region having most pronounced differences between M0 and M2 phenotypes.

**Figure 4 ijms-24-00824-f004:**
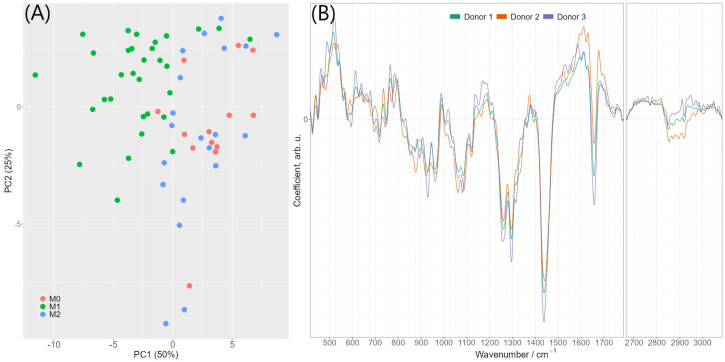
(**A**) 2D-PCA scores plot of the first two principal components. Scores were calculated by applying PCA to all data used for discrimination (i.e., without cross-validation) (**B**) PCA-LDA coefficients of the model differentiating between M1 phenotype and M0/M2 phenotypes using two principal components. The colors indicate different cross-validation iterations (olive green—donor 1 was in testing set, orange—donor 2 in testing set, violet—donor 3 in testing set). The image also shows that values are relatively stable.

## Data Availability

Data is available from the authors upon request.

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
