# Peer review of "Label-Free Characterization of Macrophage Polarization Using Raman Spectroscopy†"

_ijms, 2023, doi:10.3390/ijms24010824_

Round 1

Reviewer 1 Report

This manuscript demonstrates that Raman spectroscopy can identify differences in the biochemistries of primary M0, M1, and M2 macrophages, and that M1 cells may be discriminated from the other two macrophage phenotypes by PCA-LDA of the Raman spectra.  A major strength of the manuscript is that biochemical information about the cell biochemistries were extracted from the spectra, and the accuracy of this information was verified by comparison to the results of prior studies. Another strength was the clarity of the introduction, which clearly explained the cells’ biologies and the importance of the question that was addressed in the manuscript.  The major weakness of the manuscript is the brevity of the descriptions of the multivariate analysis techniques that were used in this manuscript, which left the reader with several questions. The reader also questioned why the authors analyzed chemically fixed cells after saying multiple times that the strength of the Raman analysis is that it is non-invasive.  These two weaknesses and the questions associated with them are described with more detail in the list below. Though these weaknesses are minor, addressing them would improve the manuscript.

11.  How does the N-FINDR algorithm that was used in this manuscript work?  Does it find (linear?) combinations of spectral features that are characteristic of a property of interest, such as biomolecule class? If so, how does this algorithm identify features related to biochemistry?  Is it a supervised technique or unsupervised technique?  Does this method use a spectral database of different biomolecules for comparison?  Or are the spectral features identified according to spatial distribution?   Why did the authors use N-FINDR instead of PCA for spectral images?  The addition of a brief description of how this algorithm works that answers these questions into the results section would help broaden the potential readership of the manuscript.  In a related question, what is meant by the term “abundance threshold”, which was used on page 4 line 168. Is that some sort of threshold that is used for identification?

22. The phrase, “stable quality of 85 – 90%” on page 6 line 239 is not clear. An explanation of what the percentage range refer to is needed. In addition, what does “stable quality” mean?  Does this phrase refer to the reproducibility of identifying the cells that belong to the macrophage phenotype of interest?

33. Section 3.3 states that 15 M0, 31 M1 and 20 M2 cells were analyzed.  Please state how many M0 cells were from donor 1, 2, and 3; how many M1 cells were from donor 1, 2, and 3, and how many M2 cells were from donor 1, 2 and 3.

44. Finally, the abstract, introduction and conclusion all state that the 3 macrophage phenotypes were accurately discriminated in a non-invasive, non-destructive and label-free manner.  Use of non-invasive and non-destructive gives the impression that the cells were alive when they were analyzed, when the methods section indicates they were chemically fixed. Though I agree that Raman spectroscopy can in theory be performed on living cells without harming them, I assume that something prevented the authors from doing so. To avoid misleading the reader, claims of non-invasive and non-destructive identification should be removed, the potential for performing these identifications non-invasively and non-destructively could be emphasized instead, and an explanation for why the measurements were performed on fixed cells should be provided.

Author Response

This manuscript demonstrates that Raman spectroscopy can identify differences in the biochemistries of primary M0, M1, and M2 macrophages, and that M1 cells may be discriminated from the other two macrophage phenotypes by PCA-LDA of the Raman spectra. A major strength of the manuscript is that biochemical information about the cell biochemistries were extracted from the spectra, and the accuracy of this information was verified by comparison to the results of prior studies. Another strength was the clarity of the introduction, which clearly explained the cells’ biologies and the importance of the question that was addressed in the manuscript. The major weakness of the manuscript is the brevity of the descriptions of the multivariate analysis techniques that were used in this manuscript, which left the reader with several questions. The reader also questioned why the authors analyzed chemically fixed cells after saying multiple times that the strength of the Raman analysis is that it is non-invasive. These two weaknesses and the questions associated with them are described with more detail in the list below. Though these weaknesses are minor, addressing them would improve the manuscript.

  1. How does the N-FINDR algorithm that was used in this manuscript work? Does it find (linear?) combinations of spectral features that are characteristic of a property of interest, such as biomolecule class? If so, how does this algorithm identify features related to biochemistry? Is it a supervised technique or unsupervised technique?  Does this method use a spectral database of different biomolecules for comparison?  Or are the spectral features identified according to spatial distribution?   Why did the authors use N-FINDR instead of PCA for spectral images?  The addition of a brief description of how this algorithm works that answers these questions into the results section would help broaden the potential readership of the manuscript.  In a related question, what is meant by the term “abundance threshold”, which was used on page 4 line 168. Is that some sort of threshold that is used for identification?

Answer to question 1: We have now added more details of the algorithm in the “Materials and Methods” section. To not confuse a reader with too much of technical details, we keep it short and gave reference to sources where more details can be found. In addition, we provide a brief answer to the main questions of the reviewer below.

1a) How does the N-FINDR algorithm work

N-FINDR algorithm was first published by M. Winter (N-findr: An algorithm for fast autonomous spectral end-member determination in hyperspectral data. SPIE: 1999; Vol. 3753). It is one linear spectral unmixing algorithm out of a larger group of techniques for representing data as a mixture (linear combination) of pure components unknown beforehand. Using formula, the problem of N-FINDR can be described as following. Generally, the spectra for a given pixel in an image is assumed to be a linear combination of the endmember spectra:

zi = c1 e1 + c2 e2 + ... + cp ep + epsilon, ci >= 0,  Sum ci = 1 (eq.1)

(see formula in attached pdf)

      Where ei are unknown beforehand pure component (endmember) spectra, ci are the corresponding coefficients/concentrations, and  is gaussian random error (assumed to be small). So, the main focus of the N-FINDR is to find endmember spectra for a given number of pure components p. The algorithm is based on the fact that, if for each pure component there is at least one spectrum in the data, the points meeting Eq.1 requirement would form a simplex (a generalization of the notion of a triangle or tetrahedron to arbitrary dimensions) in p-1 space. Therefore, the simplex formed by endmembers would have the largest volume (comparing to simplexes formed of non-pure pixels), i.e. the endmembers can be found by finding p spectra within data set that span (p-1)-simplex with largest volume.

1b) Assignment to biomolecule class

From the description above we can say that it is linear, unsupervised, and it does not use any external databases. The assignment to certain biomolecule class (nuclei, lipids, proteins) is based on both visual inspection of the image and on spectral features (i.e. Raman bands) of the endmembers. I.e. it is done manually by carefully checking each endmember.

1c) Why use N-FINDR instead of PCA

Indeed, both algorithms are related in many ways. PCA used in N-FINDR for initial dimension reduction. Also, both techniques result in linear decomposition of the spectral data. However, the original motivation for the algorithms is different: N-FINDR was developed for finding pure components (endmembers) while the idea of PCA is to reduce the data dimensionality by keeping as much information as possible in orthogonal way. The main advantages are that the endmembers (in contrast to PCA loadings) are easy interpretable Raman spectra that can be tracked back to a specific pixel in the corresponding Raman image; and the decomposition coefficients (in contrast to PCA scores) are non-negative values interpreted as estimated relative concentrations.

1d) Abundance thresholding

After the endmembers are found, each pixel in the image is described as a linear combination of those endmembers (see also equation 1 above). The coefficients ci describe the relative abundance of i-th endmember for each pixel. If one is interested in the distribution of a particular endmember or wants to extract the original Raman spectra with high contributions of a particular endmember, one could define a threshold value of the relative abundance ci, to only select spectra which contain the endmember of interest. This approach was used for selecting the lipids region (line 168 on page 4 referred to that). In the revised version of the manuscript, we have rephrased the description and moved details of selecting lipid spectra to the figure caption of the corresponding supplemental Figure S6-B referenced in the manuscript.

  1. The phrase, “stable quality of 85 – 90%” on page 6 line 239 is not clear. An explanation of what the percentage range refer to is needed. In addition, what does “stable quality” mean? Does this phrase refer to the reproducibility of identifying the cells that belong to the macrophage phenotype of interest?

Answer to question 2: Indeed, the phrase appears to be ambiguous. The corresponding part of the manuscript was adopted to represent results clearer. The range was replaced by providing exact number for each metric used, i.e. balanced accuracy 86%, sensitivity 93%, specificity 85%, and ROC AUC 93%. And the phrase “stable quality” was removed.

  1. Section 3.3 states that 15 M0, 31 M1 and 20 M2 cells were analyzed. Please state how many M0 cells were from donor 1, 2, and 3; how many M1 cells were from donor 1, 2, and 3, and how many M2 cells were from donor 1, 2 and 3.

Answer to question 3: This information is given in Supplementary Table S1. The reference to this Table is given at the end of the sentence:
In section 3.3 it now reads: “In total, 67 cells (15 M0, 32 M1 and 20 M2) were measured yielding in 506,246 spectra (Detailed split per donor is given in Supplementary Table S1)”.

  1. Finally, the abstract, introduction and conclusion all state that the 3 macrophage phenotypes were accurately discriminated in a non-invasive, non-destructive and label-free manner. Use of non-invasive and non-destructive gives the impression that the cells were alive when they were analyzed, when the methods section indicates they were chemically fixed. Though I agree that Raman spectroscopy can in theory be performed on living cells without harming them, I assume that something prevented the authors from doing so. To avoid misleading the reader, claims of non-invasive and non-destructive identification should be removed, the potential for performing these identifications non-invasively and non-destructively could be emphasized instead, and an explanation for why the measurements were performed on fixed cells should be provided.

Answer to question 4: We have taken the suggestion of the reviewer into account and added the information that we were using chemically fixed cells in the abstract, directly in the result section (section 2.2) as well as in the conclusion section in order to avoid any misconception. We have also added a brief explanation in experimental section 3.3 why we fixed the cells in this study and mentioned in the conclusion that measurements can also be done with living (non-fixed) cells.

In our opinion, the chemical fixation belongs to the sample preparation step, while the general method (i.e., Raman spectroscopy) is non-invasive and non-destructive. Thus, we kept the statement that we analysed the (chemically fixed) cells non-invasively and non-destructively. Here, we rely on general definitions of the terms: The NIH defines “non-invasive procedures” as those that “do not involve tools that break the skin or physically enter the body. Examples include x-rays, a standard eye exam, CT scan, MRI, ECG, and Holter monitoring”. This definition is fulfilled when using Raman spectroscopy (independent of sample preparation, i.e. independent if the cells are investigated alive or chemically fixed.). Also, as during Raman spectroscopy “no destruction of material being investigated or treated” is occurring, it is a non-destructive technique. That is also in agreement with the use of the terms in current publications in the research field. If living cells are investigated, it is usually stated additionally in the title (as it is not yet standard procedure yet).

Reviewer 2 Report

This manuscript reports on label-free detection of macrophage using a Raman spectroscopy. Although, this manuscript could be of interest to the readers of this journal. There are a few issues that need to be addressed before suggesting for publication.

-          The introduction needs to be strengthened by providing more references on the use of label-free Raman and what are the techniques to enhance Raman signal such as  doi.org/10.1021/acsami.8b10590 and doi.org/10.1016/j.snb.2020.128703

-          Could the authors include the brightfield images of single cells mapped using Raman microscope in figure.2?

-          Could the authors include the 2D PCA plot in the main manuscript and elaborate more on the discrimination algorithm used ?

Author Response

This manuscript reports on label-free detection of macrophage using a Raman spectroscopy. Although, this manuscript could be of interest to the readers of this journal. There are a few issues that need to be addressed before suggesting for publication.

  • The introduction needs to be strengthened by providing more references on the use of label-free Raman and what are the techniques to enhance Raman signal such as doi.org/10.1021/acsami.8b10590 and doi.org/10.1016/j.snb.2020.128703

Answer to question 1: We have taken the suggestion of the reviewer into account and added the following references:

[15] Hobro, A.J.; Smith, N.I. Label-free Raman imaging. In Nanotechnology characterization tools for biosensing and medical diagnosis, Kumar, C.S.S.R., Ed. Springer Berlin Heidelberg: Berlin, Heidelberg, 2018; pp 277-331.

[20] Hobro, A.J.; Kumagai, Y.; Akira, S.; Smith, N.I. Raman spectroscopy as a tool for label-free lymphocyte cell line discrimination. Analyst 2016, 141, 3756-3764.

We have not added references with (any) enhanced Raman techniques as we did not apply those techniques and therefore think it would be beyond the scope of the manuscript.

  • Could the authors include the brightfield images of single cells mapped using Raman microscope in figure 2?

Answer to question 2: We now also show brightfield images of the single cells of which false color Raman images are depicted in Figure 2. Those images are now included as Figure S4-E in the Supplement.

  • Could the authors include the 2D PCA plot in the main manuscript and elaborate more on the discrimination algorithm used?

Answer to question 3: The suggestion of the reviewer was taken into account. The manuscript was updated accordingly. A 2D PCA scores plot of the first two principal components was added to the Figure 4 as panel A. We have also added an additional comment and reference to the method’s section to explain our choice of algorithm. Details on the used algorithm are provided in the cited work (references 47 and 48 of the revised manuscript) as well as in the newly inserted reference (Gautam, R.; Vanga, S.; Ariese, F.; Umapathy, S. Review of multidimensional data processing approaches for Raman and infrared spectroscopy. EPJ Techniques and Instrumentation 2015, 2, 8, doi:10.1140/epjti/s40485-015-0018-6).